# Sustainable Ship Management Post COVID-19 with In-Ship Positioning Services

**Qianfeng Lin** [1] and **Jooyoung Son** [2,*]

1   Department of Computer Engineering, Korea Maritime and Ocean University, Busan 49112, Korea; linqianfeng@g.kmou.ac.kr
2   Division of Marine IT Engineering, Korea Maritime and Ocean University, Busan 49112, Korea
*   Correspondence: mmlab@kmou.ac.kr

**Abstract:** COVID-19 is spreading out in the world now. Passenger ships such as cruise ships are very critical in this situation. Boats' hazardous areas need to be identified in advance and managed carefully to prevent the virus. Therefore, this paper proposes for the first time that three technologies are required to support the sustainable management of ships in the post-COVID-19 era. They are ship indoor positioning, close contact identification, and risk area calculation. Ship environment-aware indoor positioning algorithms are proposed for the first time for the moving ship environment, followed by a clustering algorithm for close contact identification. Then, the risk area is calculated using the convex hull algorithm. Finally, a sustainable management approach for ships post COVID-19 is proposed.

**Keywords:** in-ship positioning; post COVID-19; close contacts; risk area; sustainable ship management





## 1. Introduction

The pandemic of COVID-19 has had a significant impact on cruise operations worldwide. This essential economic and employment activity will be restarted gradually. Cruise operators must ensure that cruises do not pose unacceptable health hazards to passengers, crew, and the general public compared to other package holidays. The shut-off spatial environment of ships facilitates the spread of the virus in a global COVID-19 pandemic environment. Cruise ships have contributed to the spread of COVID-19 worldwide [1]. The Japanese government ordered the passengers and crew on the Diamond Princess to start a two-week quarantine after a former passenger tested positive for COVID-19 [2]. Crew members are one of the occupational groups more susceptible to outbreaks of this virus due to the cramped working environment [3]. The importation of viruses onboard is facilitated by regular and irregular contact with land-based populations. Living in confined spaces with limited air exchange promotes the spread of disease [4,5].

The safe running of any cruise ship typically necessitates the participation of multiple parties, including the ship management firm. Risk management is the most crucial measure to ensure the proper operation. The work of [6] investigates the frequency, circumstances, and causes of occupational accidents on merchant ships. They conclude that occupational accidents remain a crucial issue for the shipping industry. The work of [7] identifies key points for accident prevention onboard ships, with people being critical points. Threat information collected and recorded using a structured approach can improve ship safety by selecting risk control options [8]. The work of [9] considers risk management of ship fire incidents as an essential issue in maritime transport systems. Ultimately, risk management is really about managing people. Therefore, real-time access to information on the location of persons on board is an essential element of risk management. The first stage for cruise firms is to identify any COVID-19-related dangers to their ships, crew, guests, and other people and put in place suitable precautions [10]. It is suggested that a ship management plan be made. The corporation shall evaluate any health hazards to passengers related to

the COVID-19 pandemic, crews, and passengers. The corporation should put adequate protection to minimize the risk to the greatest extent possible [11].

To manage ship personnel sustainably in epidemiological conditions, it is necessary to know their location on board. Indoor positioning technology can help ship managers see crew members' real-time location distribution. The technology can also locate the site of an incident in time should they be in danger, and rescue can be carried out quickly.

Beacon is an IoT device based on BLE (Bluetooth Low Energy) for short-range communication. It has the advantages of low power consumption, miniaturization, wide signal range, and low cost. Despite the benefits of BLE beacons, their performance is less than ideal in terms of indoor positioning accuracy when used on ships such as passenger ships [12]. Existing indoor positioning algorithms have a wide error range and can cause severe problems in narrow and complex areas inside passenger ships [13]. Therefore, there is a need to develop a new indoor positioning algorithm for beacons in the ship environment. However, the traditional indoor positioning algorithm scenario is in a fixed building, whereas a ship is a moving building. This makes conventional indoor positioning algorithms not directly applicable to the ship environment. Therefore, there is a need to investigate indoor positioning algorithms for ships [14].

In the mobile ship environment, existing positioning systems have significant application problems due to dynamic internal and external influences, such as changes in sailing speed [15]. These challenges are mainly due to the arbitrary movement of the ship and the resulting complex effects on the indoor radio signal [16]. Such products can lead to constant changes in the fingerprint profile, resulting in a degradation of positioning accuracy.

In this paper, new indoor positioning algorithms are proposed for the first time to address these challenges. The traditional fingerprint mapping localization algorithm has offline and online phases. The offline phase is unnecessary in the new indoor positioning algorithm and goes directly to the online phase. This method does not depend on the fingerprint map, which reduces the accuracy error caused by the change of the fingerprint map.

Ship indoor positioning algorithms, close contact identification algorithms, and risk area identification algorithms have been studied similarly in their respective fields. However, there are no examples of close contact identification algorithms and risk area identification algorithms studied in a ship environment. The topic presented in this paper is location services for the sustainable management of ships. The ship environment-aware indoor positioning algorithm proposed in this paper is applicable in the ship environment. Unlike traditional indoor positioning algorithms, the ship environment-aware indoor positioning algorithm does not require the creation of a fingerprint map. The nearest beacon is determined based on the peak value of the RSSI collected by the user device. The ship environment changes frequently and has a more significant impact on the fingerprint profile. Therefore, this approach gets rid of the dependence on fingerprint maps. Thereby the algorithm can be adapted to the ship environment.

In the post-epidemic era, identifying close contacts is essential for management. The identification of close contacts is also an application case for location services. However, the ship environment is very different from a typical indoor environment. The process and algorithm details for identifying close contacts based on location are also other. Close contact recognition algorithms need to be studied for unique environments such as ships. For now, this paper fills the research gap of close contact identification algorithms in the context of ships. In addition, it is the first time that a risk area identification algorithm has been proposed for the ship environment. During a ship's voyage, the persons on board cannot enter or leave the boat. Then in a closed environment such as a ship, the persons on board will gather in some areas. Therefore, in the post-epidemic era, these areas are at risk. However, the formation of these areas depends on the behavioral patterns of the people traveling onboard. Due to the specificity of the ship environment, it is necessary to research risk area identification algorithms in the context of ships. Overall, this paper presents the idea of sustainable management of ships with positioning services in the post-epidemic era.

For the first time, a new technical framework is proposed to make practically sustainable management. It can also be applied to different types of ships.

Ship personnel location, close contact, and risk areas are the three main elements of sustainable ship management in the post-COVID-19 era. The algorithms and techniques to achieve these three elements are shipped environment-aware indoor positioning algorithms, stop point extraction techniques, and close contact and risk area identification. This paper will focus on the above algorithms and methods and discuss sustainable ship management in the post-COVID-19 era.

This paper is organized as Section 3 introduces the ship environment-aware indoor positioning algorithm to obtain the location information of the people riding the ship and to compare other different indoor positioning algorithms. Section 4 introduces the method of mobile information extraction. Moreover, the mobile information extraction is based on the location information of the ship's crew. In addition, the concept of area of interest is introduced to calculate the risk area. Section 5 is to introduce close contact person identification and risk area identification. The close contacts are distinguished by the clustering algorithm. After the clustering results are obtained, the risk region calculation is then performed. Section 6 is a discussion on how to manage ships sustainably in the post-epidemic era based on our proposed approach.

## 2. Previous Works

The public spaces onboard are also characterized by extensive use of steel for walls, floor and ceiling, and aluminum for doors, handrails, and panels [17]. Received signal strength indication (RSSI) is affected by the steel structure and dynamics of the shipboard environment. This is because changes in the ship environment are difficult to predict. For example, unpredictable dynamic factors can cause changes in the fingerprint map [18]. Such unforeseen emotional factors such as sailing speed, acceleration motion, turning motion, and weather conditions, it is inefficient and impractical to analyze and model the effects of each element. The model parameters need to be explored under all possible navigational conditions to adapt existing positioning methods in the mobile ship environment, which is time-consuming and costly system deployment [19]. The traditional indoor positioning algorithm is a fingerprint map approach that provides stable positioning accuracy and low positioning error in a stationary building. However, in the case of ship navigation, the fingerprint map changes according to different dynamic factors. Once the changes occur, the fingerprint map needs to be remeasured. This approach in the mobile ship environment leads to low efficiency and high error. Therefore, this paper proposes a new algorithm that does not rely on the fingerprint map in the offline phase. The localization is performed directly in the online phase, and high accuracy localization results can be obtained. The new indoor positioning is called ship environment-aware indoor positioning algorithms.

The location of the ship personnel is a crucial piece of data for the discovery of close contacts. The most critical aspect of pandemic preparation is the timely detection of relative references on board. The location of smartphones can identify close contacts. Nevertheless, the environment is challenging to predict. Such unpredictable dynamic factors such as sailing speed, acceleration motion, turning motion, weather conditions, and so on [20]. It is inefficient and impractical to analyze and model the effects of each factor. The model parameters need to be explored under all possible navigational conditions to adapt existing positioning methods in the mobile ship environment, which is time-consuming and costly system deployment [21].

Most indoor positioning algorithms using infrastructure have two-phase processes: offline and online. In the offline phase, a fingerprint map is made with peak or average RSSI values gathered at RPs (reference points). In the online phase, a user position is estimated. The nearest K RPs are looked up in the fingerprint map using the peak or average RSSI values of beacon signals received by a user device. The Euclidean distance among the K RPs is calculated to estimate the user location.

The clustering algorithm can detect close contacts based on their location similarity to COIVD 19 patients. The COVID-19 patient and close contacts are observed as one group. Once close contacts have been identified, they can be isolated, thus interrupting the spread of the virus. For the sustainable management of ships, detecting close contacts and identifying risk areas is necessary. After obtaining the location of close contacts, the area where they stay can be accurately calculated. The convex hull algorithm is the main algorithm to implement the calculation of risk areas.

The density-based spatial clustering (DBSCAN) algorithm is one of the clustering algorithms. DBSCAN is now widely used in various fields [22]. DBSCAN algorithm selects the outliers from each cluster, and K-means clusters are divided into cluster-self and sub-cluster. The authors of [23] propose a directional-DBSCAN line-level feature-clustering algorithm and slot detection with slot pattern recognition. The authors of [24] use the DBSCAN clustering algorithm to extract similar ship trajectories in state space to calculate the behavior patterns of different ships. DBSCAN, which measures similarity based on distance, is an excellent way to distinguish close contacts.

Close contacts and confirmed patients with COVID-19 usually stay in one area for an extended period. The location points where they stay are called dwell points. These dwell points are referred to as OD (origin-destination) data. Similarly, OD data are used in many fields. The authors of [25] estimate OD flows using opportunistically collected mobile phone location data from one million users in the Boston metropolitan area. The authors of [26] note that a stop detection algorithm derives the user activity locations and times based on the raw data collected by their phones. The OD data indicates the geographical space shift from origin to destination. For this paper, OD data is critical for calculating risk areas.

The commander's area of interest is called the area of interest (AOI). The risk area is one of the AOI. The convex hull algorithm is the key algorithm to calculate the risk area. The convex hull algorithm is also used in several fields. The authors of [27] show that a convex hull algorithm can determine the boundary nodes among a set of nodes in the network. In this paper, after obtaining the OD data, it is possible to calculate the risk area using the convex hull algorithm.

The approach proposed in this paper consists of three main techniques. They are the ship indoor positioning algorithm, close contactor identification, and risk area identification. To validate the effectiveness of the proposed method, the location information of the user devices needs to be collected firstly. A deployment plan needs to be made based on the ship's floor plan to deploy the beacon devices accurately and efficiently. In the method proposed in this paper, the user device refers to a smartphone. Firstly, the RSSI data from the beacon needs to be collected using the smartphone's app. Secondly, the RSSI data is fed into the ship environment-aware indoor positioning algorithm to obtain the location information of the user device. The exact positioning process has been discussed in Section 3. In the deployment planning, test locations for close contacts are pre-set. These test locations are located in different areas. There will be two or three test locations in different zones. In each area, the distance between the test locations is less than 2 m. At each test location, 15 min of RSSI data is collected using different smartphones.

To simulate a realistic scenario onboard the ship, a person on board with a smartphone will walk through the area where these test locations are located. The test positions are also estimated using the ship environment-aware indoor positioning algorithm. The resulting result of location estimation is fed into the close contact identification algorithm, which returns a clustering result. The categories to which the user IDs of close contacts and COVID-19 confirmers belong in the deployment planning are known. The estimated results can be compared to the known results to verify that the close contact identification algorithm is valid. Finally, the risk area identification algorithm is used to calculate the area where the estimated test location is located. Since the area in which the test location is located is known in the deployment planning. Therefore, the similarities and differences

between the estimated size and the known area can be compared to verify the validity of the risk area identification algorithm.

The uncertainty about the proposed method mainly comes from the localization error. Traditional indoor localization algorithms use fingerprint maps. However, in a complex environment, the fingerprint map is constantly changing. Fingerprint map needs to be recreated at regular intervals, which is difficult to achieve in a mobile environment such as a ship. Therefore, the ship environment-aware indoor positioning algorithm proposed in this paper does not rely on a fingerprint map for positioning. Thus, the uncertainty of the proposed method is reduced.

This paper focuses on three major elements of sustainable ship management post COVID-19, in order of ship environment-aware indoor positioning algorithms, OD data extraction, AOI identification, DBSCAN algorithm, and risk area identification. Finally, the post-COVID-19 ship sustainable management model discusses early quarantine measures and rapid action after discovering confirmed patients.

## 3. Ship Environment-Aware Indoor Positioning Algorithms

The test environment is set up on the HANNARA ship, a Korea Maritime and Ocean University training ship. In Figure 1, the red dot symbols stand for the positions of beacons, and the green dot symbols mean the RPs that are precisely determined and measured in advance. The marks (1)~(25) are arbitrarily chosen user locations at which the algorithm in this paper is applied to estimate the user position. The space between beacons is 2.5 m, and between RPs is 2 m.

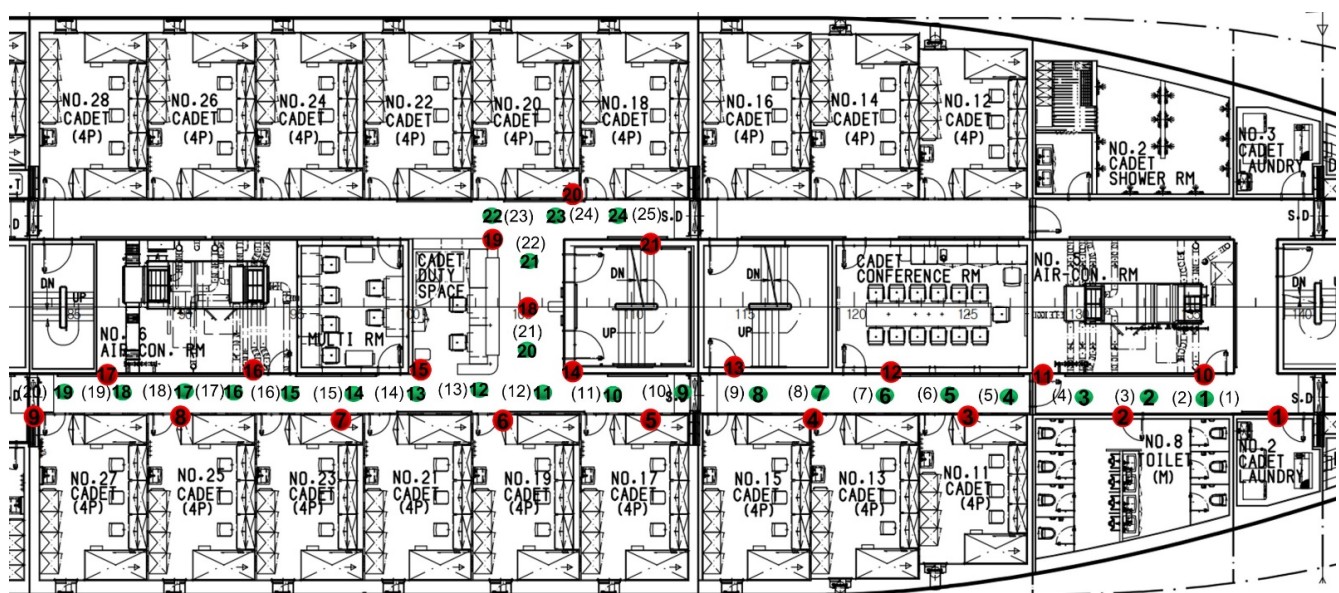

**Figure 1.** The experimental environment in HANNARA ship.

This paper proposes an algorithm to adapt to the complex environment of ships. The algorithm considers the nearest beacon whose signals are received by the user device in the online phase. The ship environment-aware algorithm is based on the peak value of RSSI to determine the distance between the two nearest beacons to the user. Since the position of the RP is known, the two RPs can be found if the two beacons are identified, and the user location can be estimated from the position of the nearest beacon and the position of the two RPs. However, considering that the user may be at the edge of the beacon coverage, the RP closest to the beacon is estimated as the user location when the nearest beacon is known.

Since it is necessary to calculate the location relationship between beacons and RPs, in steps 1–2, the coordinates of beacons and RPs can be stored as arrays C and R. Because

beacons have limited coverage and often produce significant errors at the covered boundaries. The algorithm should recognize this situation. In step 3, the beacon IDs of boundaries should be stored as list B. In step 4, the correspondence of the nearest RP to each beacon with arrays B and C are determined as dictionary D. The key of D is the beacon ID. The value of D is the RP ID. In step 5, the RSSI is received from the user device. In step 6, the peak value of RSSI data in beacons is selected. In step 7, each beacon ID is sorted in descending order according to its peak value. The sequence is called S. In step 8, the first-ranked beacon ID is b1. The second-ranked beacon ID is b2. In step 9, whether b1 exists in list B. If it exists, step 10 is executed. Otherwise, step 11 is performed. In step 10, the corresponding nearest rp1 in dictionary D is found based on the beacon ID. The rp1 is estimated as the user location. In step 11, the nearest rp2 of b2 in the dictionary is chosen. In step 12, the centroid of rp1 and rp2 is calculated as P1. In step 13, the coordinates of b1 in list C are called P2. In step 14, the centroid of P1 and P2 is estimated as the user location. The specific algorithm flow is shown in Figure 2.

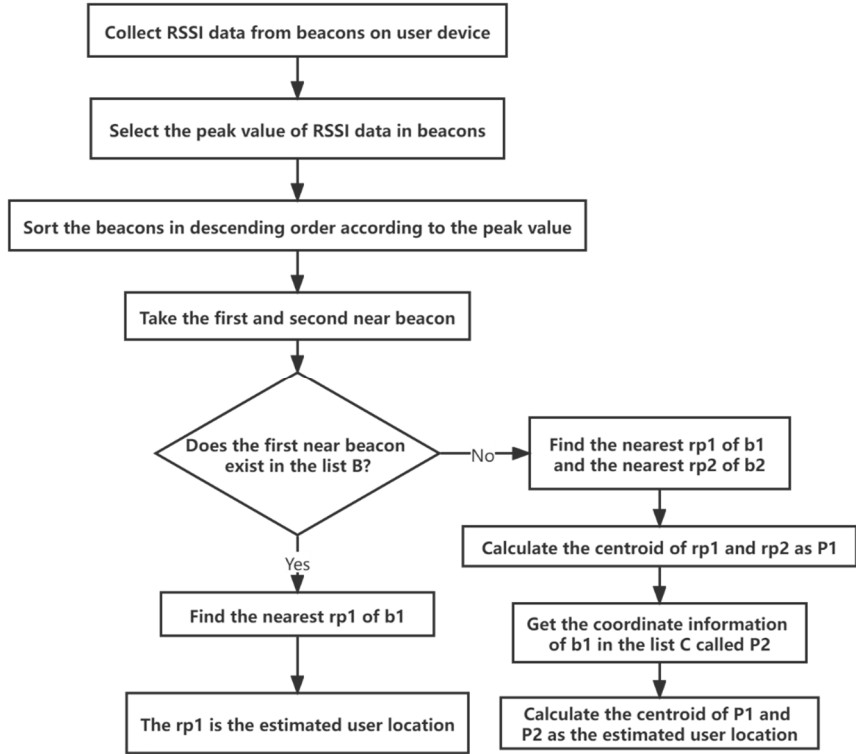

**Figure 2.** A ship environment-aware indoor positioning algorithm.

The core of the ship environment-aware indoor positioning algorithm is to find out the nearest beacon from the RSSI received from the user device. Therefore, finding a representative value from the RSSI is necessary to determine which beacon is closest to the user device. To determine which RSSI representative value gives the best results, three different RSSI representatives are input into the algorithm, namely average, K-means and peak, with root mean square error (RMSE) as the error measure. The estimated errors of the 25 user locations are shown in Figure 3. By comparison, the average case has the most significant error with an RMSE of 2.08 m. K-means is used to find out different clusters by clustering, then determine the cluster that contains the most RSSI data, and find the center of this class as the expected value of RSSI, which has the RMSE of 1.77 m.

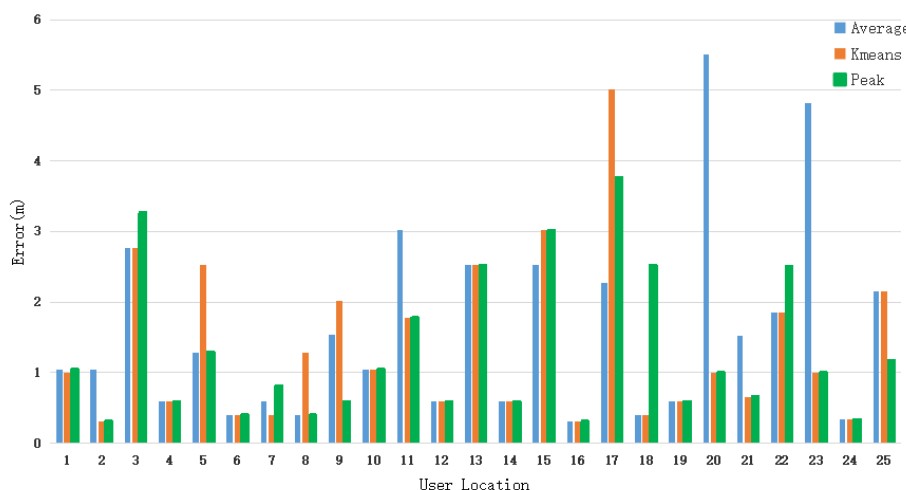

**Figure 3.** The error of each estimated user location.

The best result is the peak case with an error of 1.63 m. In general, the closer the user device is to the beacon, the stronger the RSSI. However, the complex environment of the ship can cause such a pattern to change. For example, the multi-path effect, which causes the RSSI strength to become stronger at distant places, is not in line with the standard rule. This also leads to errors in many indoor positioning algorithms, but such errors cannot be eliminated, only be continuously reduced. In the case of HANNARA, three different RSSI representations are compared in such a complex environment, and the peak case is the one that minimizes the error. Therefore, the peak value is used as the RSSI representative value in the ship environment-aware indoor positioning algorithm.

This paper compares the proposed algorithm with two other indoor localization algorithms. They are KNN and key point algorithms. Both algorithms use a fingerprint map to find the nearest RP. Still, the key point algorithm considers the nearest beacon location and estimates the user location with minor errors than KNN.

The ship environment-aware algorithm determines the nearest beacon to the user device based on the user RSSI. It determines the nearest RP to the user device based on the known relationship between the beacon and the RP. By reducing the uncertainty error through available information, the ship environment-aware algorithm shows better results in Figure 4. The RMSE of KNN, key point, and the ship environment-aware algorithms are 2.04, 1.79, and 1.63 m, respectively. The ship environment-aware algorithm has the highest accuracy and shows the best performance in complex ship environments.

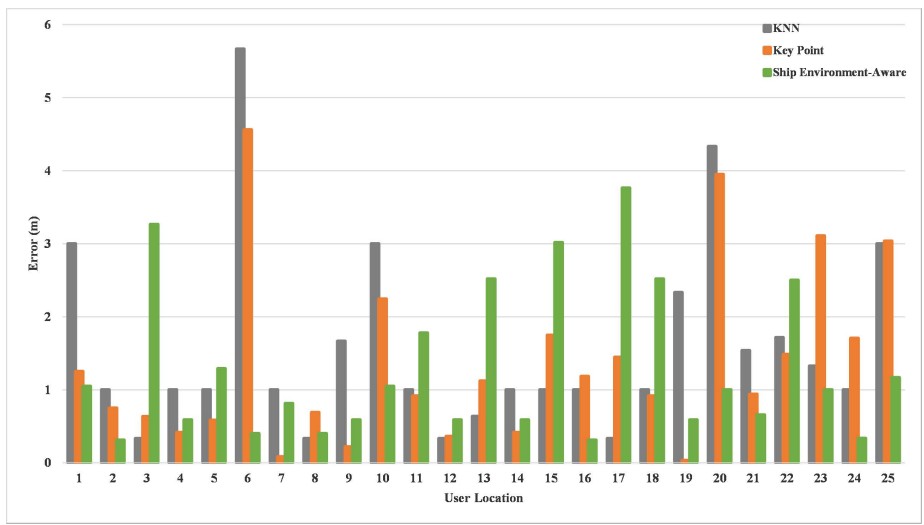

**Figure 4.** Comparison of error results of different indoor positioning algorithms.

## 4. Movement Information Extraction Technology

### 4.1. OD Data Extraction

OD data is the data of two stop points in a time position sequence. Geospatial movement information from the origin (O) to destination (D) is movement information that satisfies specific conditions and contains non-geographic details such as the number of movements. To improve persons' safety and public health on board, it is necessary to have data management technology that extracts OD data from specific conditions, such as considering risk factors. It is possible to identify the stopping point by the time and speed of movement and extract OD data based on this (Figure 5) [28].

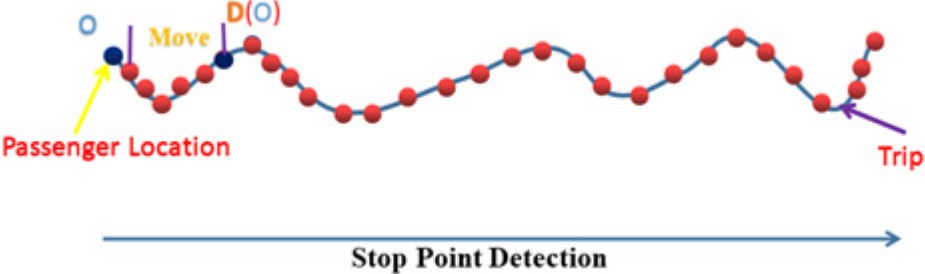

**Figure 5.** Exploring the stop point.

The OD data set, sometimes referred to as 'flow data' contains details of movement between two geographic points or, more generally, movement between zones (Figure 6) [28]. OD data sets typically have several non-geographic properties. This may include the number of trips from the origin to the destination for at least a specific time. To improve the safety and public hygiene of personnel (sailors and passengers) onboard the ship, data mining technology is essential to extract OD data that considers all specific conditions such as errors and risk factors accumulated in location information collected.

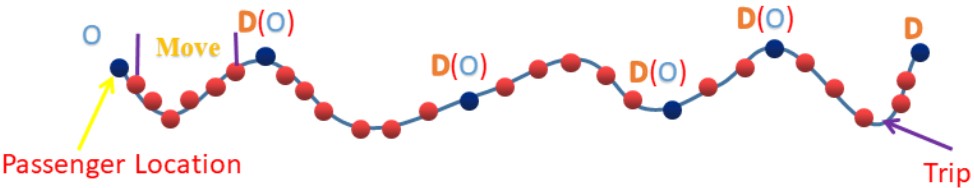

**Figure 6.** Example of OD data.

Individual OD data identified as a time series can be used. It identifies a sub-network centered on a specific node (individual) among the overall OD data represented by a complex graph. In construction, an inference technique for finding a sub-network with a contact surface should be developed. It is necessary to create a technology that clusters according to specific conditions with OD data passing through a particular area (point) on board and a data processing technology that considers location-based application services that use it in the future. For the safety of large ships, technology is required to evaluate the risk of individual boarding personnel or identified groups in specific situations using movement information. Technology for identifying an onboard risk area according to a particular risk factor based on the identified cluster information. Technology for ranking identified areas based on risk.

A location-based service model for safety and public hygiene of the number of persons on board is modeled after cruise ships to show examples of using precise location information, OD data, and identified clusters and areas. It can be used as primary data for early detection and rescue of accidents such as fainting due to high heat in engine rooms, falls in vertical structures, and being trapped in closed areas due to failure of construction devices. Technology is also required to extract OD data around a specific number of people

or a specific area and to understand the association between multiple OD data. This is expected to enhance the public hygiene of the number of persons on board, such as early quarantine of infectious diseases. This paper will calculate the risk area by combining OD data with the convex hull algorithm.

*4.2. AOI Identification*

The risk area is one of the AOI. AOI includes areas of influence, adjacent areas, and areas extending to enemy territory to current or planned operational goals. These areas also include areas occupied by enemy forces that can jeopardize mission completion. On cruise ships, rooms, snack bars, rest areas, and coasts can be set as AOI. Figure 7 shows AOI 1 and AOI 2 on a cruise ship. AOI 1 is a rest area, and AOI 2 is a room [28].

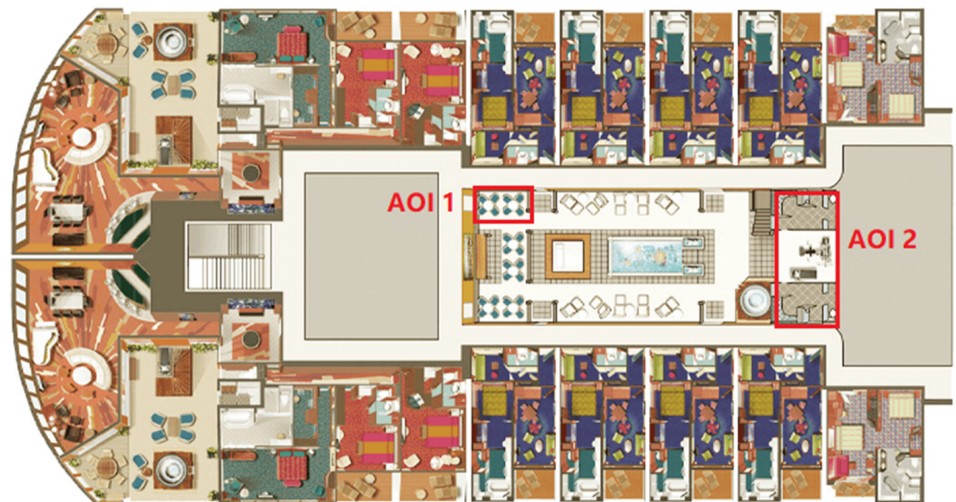

**Figure 7.** Example of AOI 1 and AOI 2 on a cruise ship.

In computational geometry, the point-in-polygon (PIP) problem determines whether a given point in a plane is inside, outside, or at a polygon boundary. A particular case of point location problems is applied in geometric data processing such as computer graphics, vision, geographic information systems (GIS), and motion planning. It deals with points in polygon questions. The end of the polygon problem may be considered in a general iterative geometric query setting. Given a single polygon and a series of query points, the answer to each query point is quickly found. A general approach to the plane point position can be used. Therefore, AOI 1 and AOI 2 can be regarded as polygons.

A stationary point indicates that the speed is zero. The first stop point of AOI 1 is the origin, and the last stop point of AOI 2 is the destination. Figure 8 shows the user track of the cruise ship floor plan. The departure point is the first stop point of AOI 1, and the destination is the last point of AOI 2. In statistics, kernel density estimation (KDE) is a nonparametric method of estimating the probability density function of a probability variable. Kernel density estimation is a fundamental data smoothing problem in which inference about the population is made based on finite data samples (Figure 8).

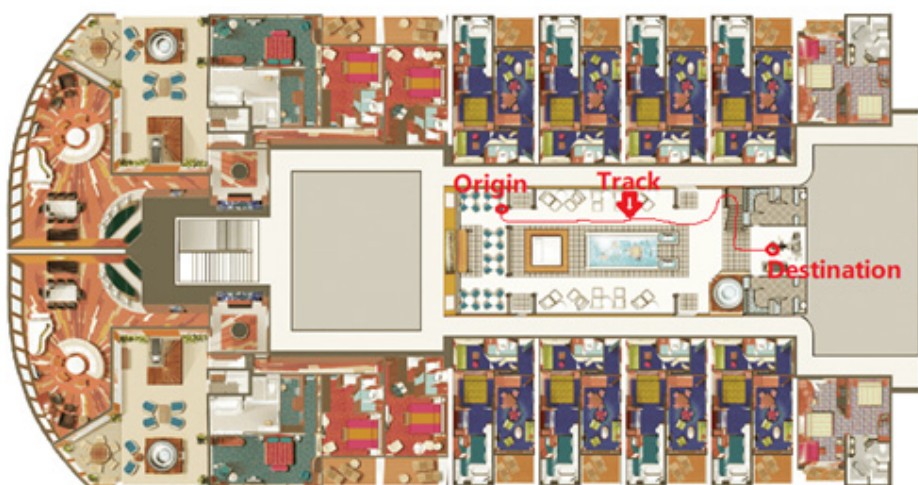

**Figure 8.** Example of departure and destination on a cruise ship.

The famous application of kernel density estimation is to estimate the class conditional limit density of data using the I Bayes classifier. This can significantly improve prediction accuracy. ArcGIS Pro is a unique and powerful desktop GIS application. ArcGIS Pro is technologically more advanced than all other products on the market, supporting data visualization, advanced analysis, and maintenance of proven data in both 2D and 3D. Hotspot area can be obtained through the origin and destination entered in the ArcGIS Pro KDE tool. Examples of results are shown in Figure 9.

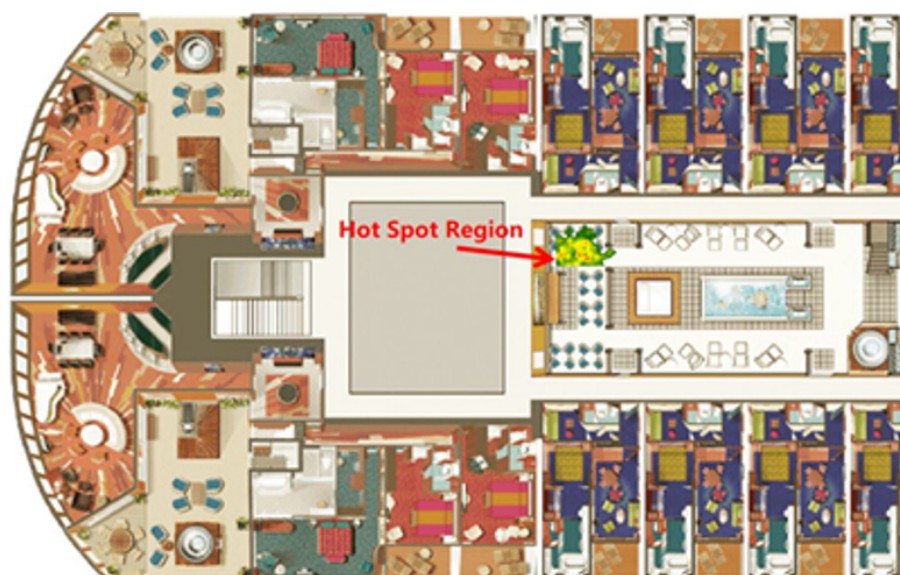

**Figure 9.** Example of the hotspot area.

## 5. Close Contact and Risk Area Identification

### 5.1. DBSCAN Algorithm

It is impossible to determine the number of close contacts and the number of clusters of confirmed COVID-19 patients in a real ship scenario. In other words, the number of infections and the number of close connections are unknown. Therefore, the classification algorithm is not suitable for application in this scenario. Clustering algorithms do not need to know the number of clusters. There are currently three different approaches to clustering data. They are partition-based methods, hierarchical methods, and density-based methods [29]. Firstly, among the partition-based methods, the most representative one

is the K-means algorithm, which requires the determination of the number of clusters or cluster centers, i.e., the pre-specification of K. The core of this algorithm is to make the points in a cluster close enough and the points between clusters far enough. Secondly, in hierarchical methods, there are two general approaches. One is agglomerative hierarchical clustering, and the other is divisive hierarchical clustering. In this example, agglomerative hierarchical clustering treats each point as a cluster, and by merging them, a cluster is obtained. However, hierarchical methods also need to specify the number of clusters.

Finally, among the density-based methods, DBSCAN is the most representative algorithm, which uses the concept of density to cluster the data. Density differentiation is most appropriate in the identification of close contacts. From a positional point of view, close contacts and confirmed COVID-19 diagnosed persons have a high similarity of characteristics in terms of density. This also means that clustering by density has high accuracy and efficiency. Not only that, but density-based clustering algorithms do not require a specified number of clusters. This makes it difficult to correctly determine K when the number of infected persons and close contacts is unknown. Both require storage of the cluster relationships between points, which poses a significant challenge in algorithmic efficiency and storage space.

Compared to other clustering methods, Figure 10 shows the comparison results of different clustering algorithms. The latitude and longitude ranges of Figure 10a to Figure 10d are the same. Figure 10a is the correct classification result. The right number of classification results is 4. The number of K-means and hierarchical algorithms clusters is directly specified as 4. It can be seen from the red box range that the clustering result of DBSCAN is closest to the correct clustering result. Therefore, DBSCAN is applied in the study of this paper.

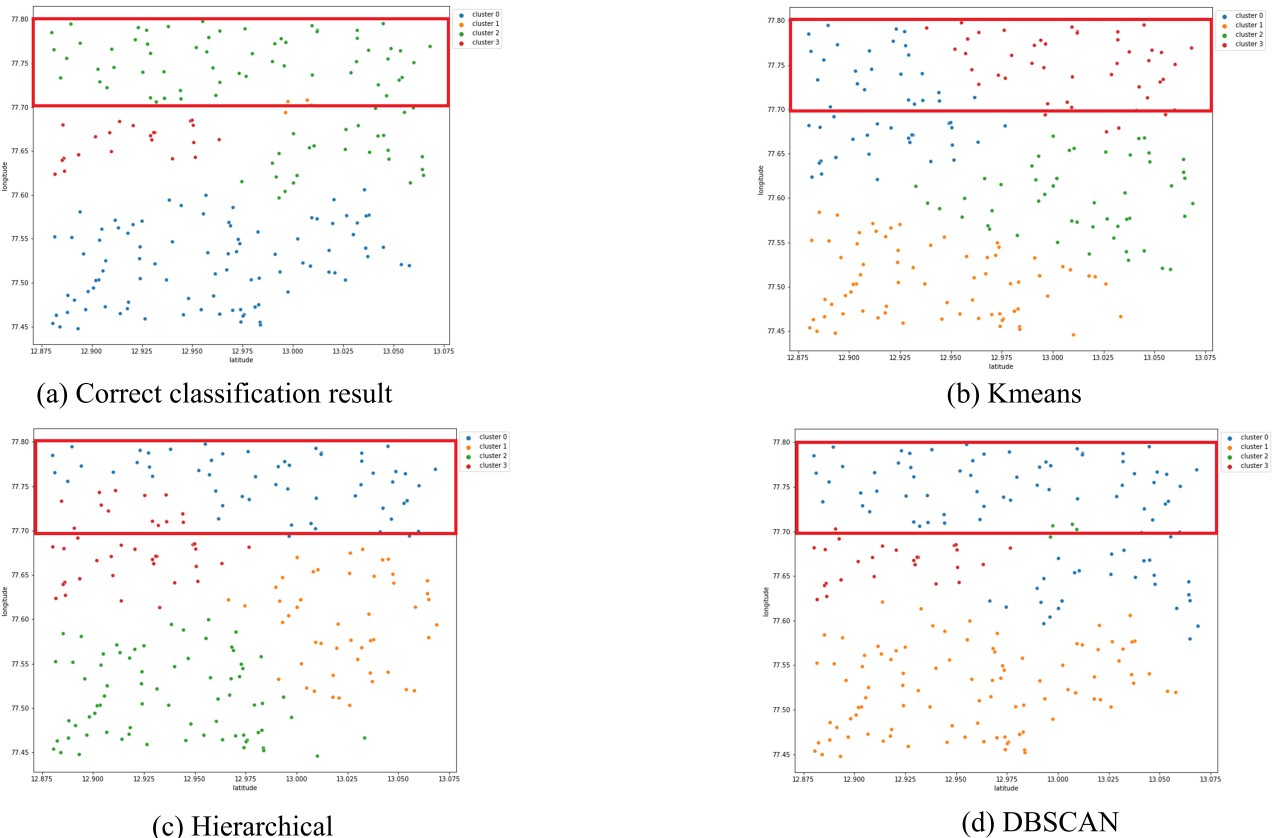

**Figure 10.** Comparison results of different clustering algorithms: (**b**) Kmeans, (**c**) Hierarchical and (**d**) DBSCASN with (**a**) correct classification result.

*5.2. Close Contact Identification*

The following section describes the entire process of finding close contact with DB-SCAN. Find all neighboring points within Eps, identify key points, or visit more neighbors than MinPts. Eps is the radius. MinPts is the minimum number of points within the radius. Density is the measure by which DBSCAN clusters the points. Therefore, density is inscribed by Eps and MinPts. In general, Eps and MinPts are initialized. The clustering results of DBSCAN are usually shown in figures. If a result shown in the figure is not suitable, the two parameters should be manually adjusted to obtain an optimal clustering result. However, this method is not suitable for use in complex ship environments. This is because the ship environment may change at any time. In the shipping scenario, the space of the ship is relatively narrow. The behavioral ways of the persons on board are very different from those in a traditional indoor method. In addition, the boat has a long navigation period. The person onboard cannot enter or leave the ship for some time, which is very different from the traditional indoor scenario. Most people live on land. As a result, there is a particular awareness of behavioral patterns indoors. This perception can help in the determination of Eps and MinPts parameters. However, most people have no experience living in ships and lack experience in the ship environment.

Faced with the many differences from traditional indoor positioning scenarios, it is difficult to determine these two parameters manually. Machine learning can provide algorithms the ability to learn the parameters. However, the data first needs to be labeled. This is very labor- and material-intensive. In addition, there is very little positional data available in the shipping scenario. There is hardly any available data to use for labeling. As in this paper, DBSCAN is used to find close contacts. There are specific behavioral characteristics between close contacts and confirmed COVID-19 cases. According to the current definition of close contact, a person is considered comparable if they spend more than 15 min with a verified COVID-19 patient at a distance of 2 m or less due to the relatively small space of the ship. Therefore, the actual contact distance is less than 2 m. However, the exact value of the spread is difficult to determine. As a result, Eps is difficult to determine.

In general, the position data of the persons on board the ship needs to be obtained every second. Assume that the acquisition time is 15 min. Then, there are 900 position data messages per person onboard. In general, both close contacts and COVID-19 patients will be in the same position for each within 15 min. Therefore, it is assumed that one of the points of a confirmed COVID-19 patient is the core point. Within a range of Eps of 2, MinPts should be 900. However, given the error of indoor positioning and the different ship environments, Eps and MinPts are set to 2 and 450, respectively. After DBSCAN has been run, the clustering results for the user ID are returned. The prevention officer can confirm the clustering results. If the number of correct results is greater than the number of incorrect results, then only the wrong results are 0, and the rest are automatically marked as 1. If the number of correct results is less than the incorrect ones, then only the correct ones are marked as one, and the rest are automatically marked as 0.

A loss function for Eps, MinPts, and labels is then constructed. This allows DBSCAN to adjust these two parameters continuously to minimize the loss value. In summary, the number of data labels is reduced by initializing Eps and MinPts. The construction of the loss function for these two parameters with the labels allows DBSCAN to adapt itself to the ship's environment and find the Eps and MinPts with the lowest loss.

Figure 11 is a visualization of the results of identifying groups under specific conditions, such as applying DBSCAN to focus on risk factors. Here, the previously extracted OD data is an essential input, but it can be added as a condition to identify groups by placing other factors. This study will also include finding appropriate input values as factors for safety.

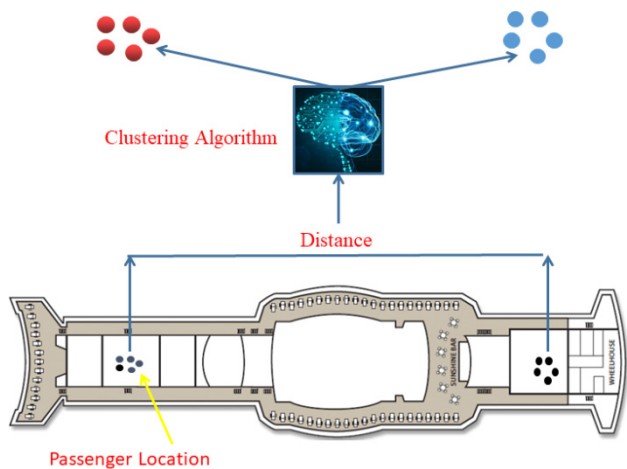

**Figure 11.** Identification of close contacts [28].

### 5.3. Convex Hull Algorithm

Algorithms that make up the convex hull of various objects have many mathematics and computer science applications. Numerous algorithms have been proposed to calculate convex hulls of finite points with varying computational complexity in computational geometry [30].

If all points are not on the same line, the convex hull is a convex polygon in which the vertex is part of the input collection box. Its most common representation is a list of vertices aligned clockwise or counterclockwise along its boundary. It is convenient to represent convex polygons as intersections of half-plane sets in some applications. From Figure 10, we know that DBSCAN classifies all location points into four clusters. Figure 12 shows that four different risk areas are calculated using the convex hull algorithm based on these four clusters of location points.

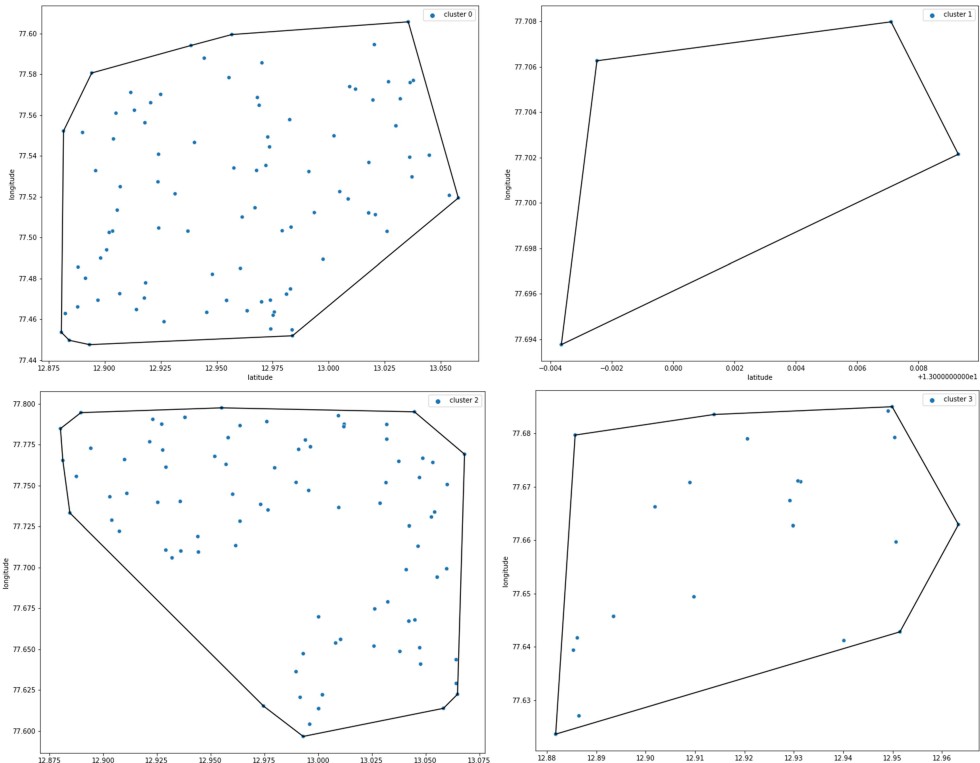

**Figure 12.** The convex hull of the point set is the smallest convex polygon, including all points.

*5.4. Risk Area Identification*

Figure 13 shows an example of identifying risk areas. DBSCAN can classify the positions of ship personnel and mark each location. Locations with the same marker are grouped into one cluster [28]. Then close contacts and COVID-19 confirmed DBSCAN classifies diagnoses. Close contacts and COVID-19 confirmed patients would be distributed in any corner on the ship. The red and blue dots represent the locations of close contacts and confirmed COVID-19 patients in different boat areas, respectively. The location data according to other clusters are input into the convex hull algorithm separately. The risk areas to different locations can be obtained. As shown in Figure 13, the red and blue risk areas are obtained, respectively.

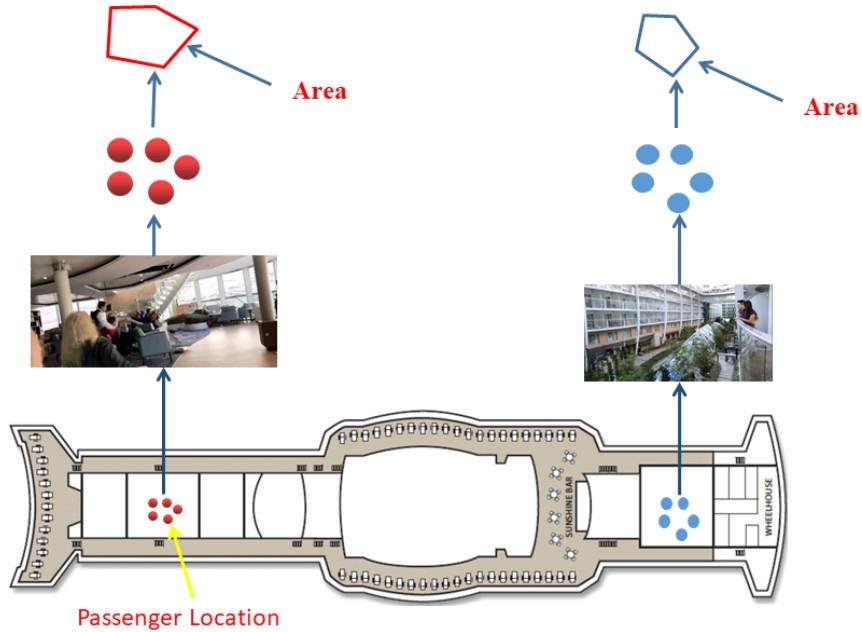

**Figure 13.** Example of identifying risk areas.

Finally, we obtain a list of zones ranked according to the degree of risk. The movement of ship personnel is one of the critical factors in the transmission of COVID-19. Suppose a patient with confirmed COVID-19 stays in a particular area for a long time. If the mobility of people in this area is high, then the risk value of this area should be very high. Therefore, the visiting rate, departure rate, and average staying time of passengers are used to measure the risk value of the area. Because the ship's personnel are constantly moving, a fixed period is needed to calculate the risk value of the area. These procedures are performed once every T-hours to immediately detect high-risk areas, feedback the results to quarantine personnel, and immediately disinfect and quarantine the dangerous areas to reduce the risk of virus transmission (Figure 14) [28]. Assuming that a potential unknown virus causes significant damage to the human body on a cruise ship, the longer people stay in a specific area, the higher the probability of being infected with the virus, and at the same time, the risk of spreading the virus increases.

This paper assumes that one of them is infected with the COVID-19 virus, which is changed into a risk area. If another person frequently enters and exits the confirmed area or stays in this area for a long time, the virus will spread quickly. These areas are usually frequently accessed by people and are characterized by staying long. The rank of the risks for each area is the purpose of our idea. Firstly, OD data are extracted using a stopping point detection method. Secondly, their residence points are classified by the DBSCAN method, and then the shape and size of the area are determined using a convex hull algorithm. Finally, three indicators of visiting rate, departure rate, and average stay time are calculated for each determined area and used to rank risks for these areas (Figure 15) [28].

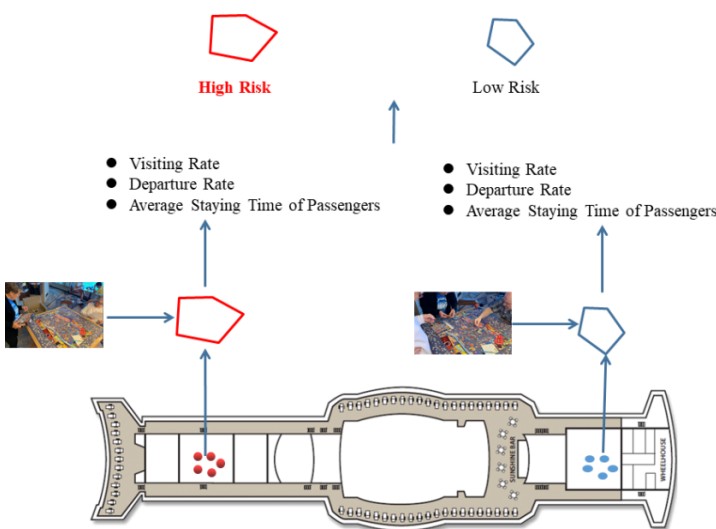

**Figure 14.** Rank the risk of spreading viruses in risk areas.

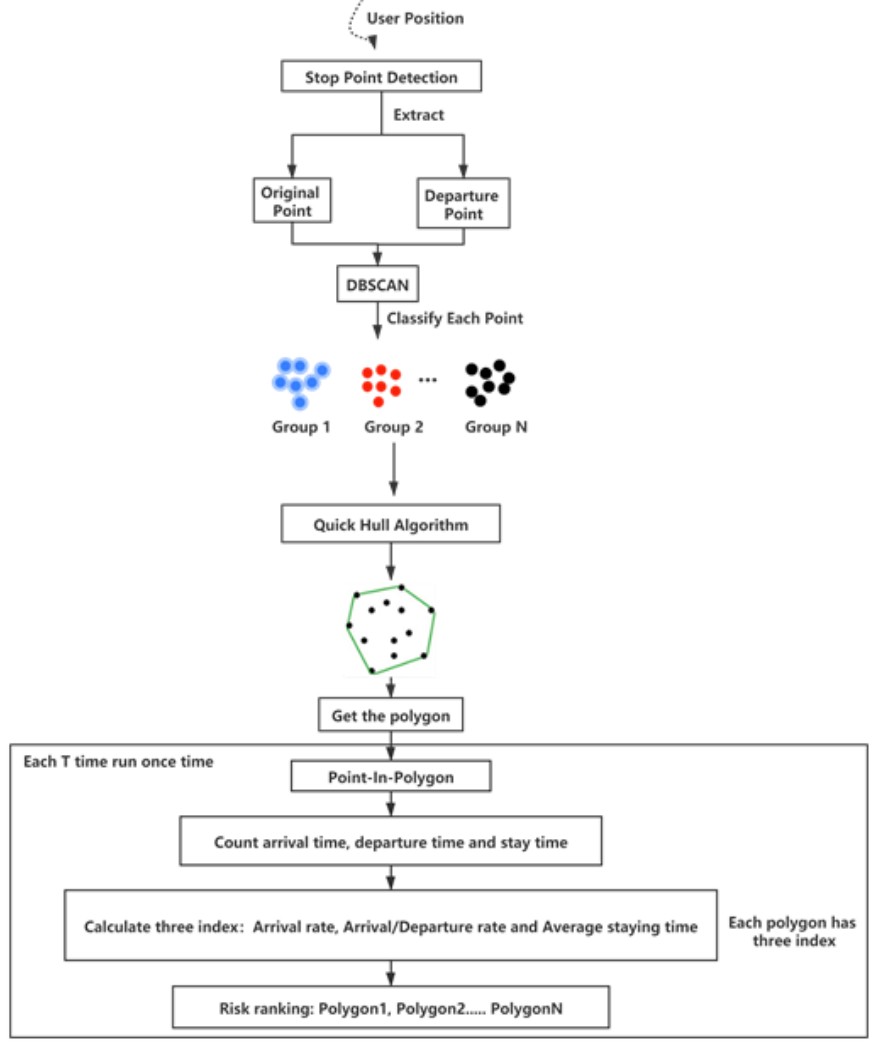

**Figure 15.** Flow chart for identifying risk areas.

## 6. Sustainable Ship Management Post COVID-19

### 6.1. Use of Development Technology: Early Quarantine Measures

Accurate passenger location information can be obtained through the ship environment-aware indoor positioning algorithm. It is possible to determine whether the physical distance between passengers is maintained with location information. If the ship passenger does not maintain a physical distance, a warning message will be sent to the passenger's mobile phone to observe the specified distance (Figure 16) [28]. Furthermore, it is judged that it may be recommended to disinfect the area by comparing the passenger location. The denser people are, the higher the risk of virus transmission. Therefore, extracting the area of stay or identifying the dangerous area can disinfect potential virus transmission areas early, reducing the risk of virus transmission. The crew identifies the passenger location information and inspects compliance with quarantine. Therefore, high precision ship indoor location tracking technology and potential viral propagation risk calculation are fundamental in early quarantine measures.

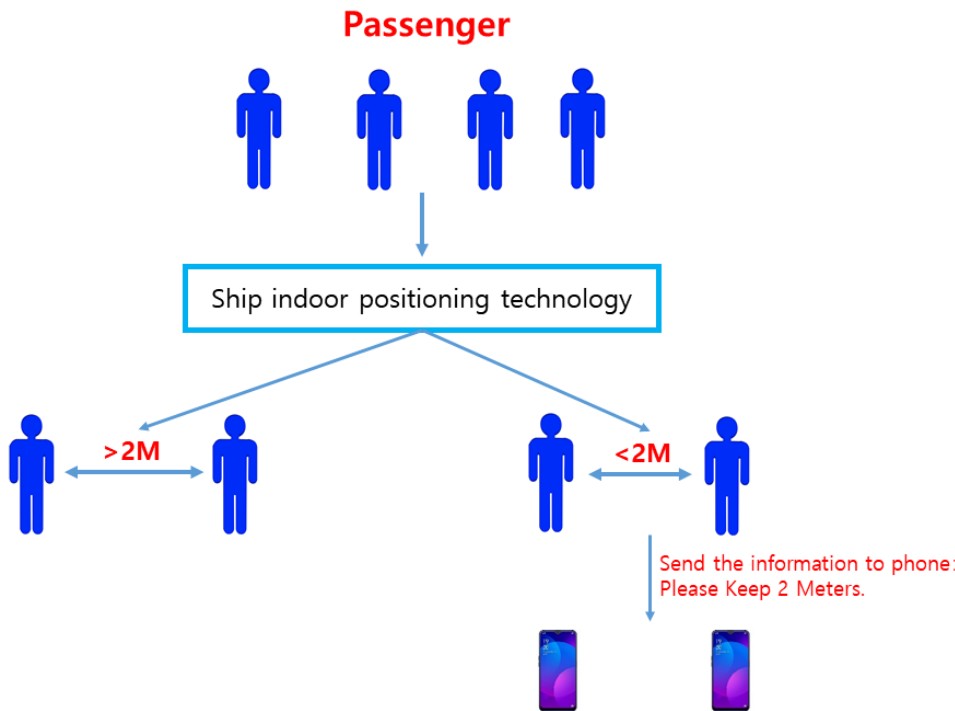

**Figure 16.** An example of applying development technology to early quarantine.

Considering the long-term stay of passengers and flight attendants, the maximum number of passengers available should be evaluated so that all necessary measures can be effectively implemented and the number can be reviewed regularly. In principle, the same level of protection should be provided to all passengers, regardless of the presence or absence of passengers, crew, or visitors. The number of passengers can be easily calculated, and trajectory information can be obtained through onboard positioning technology, allowing careful management of passengers. The onboard location identification technology can differentiate location information for each person and grade the number of persons on board by acquiring all location information for the number of persons on board. Since the number of persons on board is different, independent authority is often required in essential areas of the ship. The measures proposed in this study plan can be flexibly changed according to various application environments. For example, the following is about consumable items to be considered when including ship operation in the plan.

Consumption of personal protective equipment (PPE) may vary for each type. For example, if the number of persons onboard is only indoors, PPE is likely to be used in

duplicate. PPE may be consumed more by staff or passengers who frequently work in dense areas. This is because recycling PPE is low due to many contacts. OD data extraction technology shows that the occupant has stayed or moved. Mathematical relations with PPE consumption can predict future PPE consumption, a valuable reference for shipbuilders to plan for PPE stockpiles. Similarly, it is possible to establish a mathematical relationship between occupant movement activity and disinfectant consumption. Waste incineration can be predicted through algorithms, and disinfection equipment can be prepared in advance.

Ships can dynamically set sensory, physical distances for different onboard areas. For example, the ship engine room has a physical distance that is too small to meet quarantine regulations, so the actual physical distance for each area can be calculated considering the size of the ship space and the actual physical distance set by the number of persons on board. An electronic fence can be set up with an electronic fence through the location information of asymptomatic people. Notice can also be sent to protect the public hygiene of other passengers by guiding them to stay away from when entering closed spaces and areas with a high risk of transmission.

It is recommended that the company review where the relevant information should be provided from shipboard to disembarkation. The way information is delivered should also be reviewed and digital as much as possible. The information should include aspects related to the adopted preventive measures, the health examination process being implemented, and the protocol related to repatriation and disembarkation in the event of a disease. For example, the information to be displayed for each boarding space for the required physical distance, maximum capacity, and PPE should also be considered. The information provided should also include measures applied when visiting the community. All data may be displayed based on location information. It is to digitize the activities of onboard personnel based on location information. It can also function as a healthy constitution through the location information of the number of persons on board. For example, close contacts can be found and quarantined according to the comparison results with location information of COVID-19 patients. Ship passengers ensure that physical distance is maintained and overcrowding is prevented or reduced.

In this respect, it is vital to maintain consistency of physical distance recommendations in various areas of the ship. Floor markings representing the recommended physical distance can help keep it from passengers. If the crew interacts with a passenger in a fixed position, the protective barrier may be considered to facilitate safe interaction. If possible, removing or rearranging furniture items can reduce overcrowding. Companies should consider whether the maximum personnel capacity should be reviewed so that the physical distance applicable to each space or space category and the entire ship can be maintained. If the port and the boat use different standards, it is recommended to agree on a single distance between the port and the boat. When physical distance cannot be guaranteed, it is recommended to use a face mask as a source control means to reduce diffusion.

Based on the location information of the number of persons on board, it is possible to calculate in real-time whether the physical distance between them can meet the quarantine needs. If the requirements are not met, caution, such as sending information, can be called to maintain the physical distance of the quarantine request. In addition, when the location information of the number of persons on board is known, crowded areas can be determined or predicted through algorithms, and excessive congestion can be avoided by inducing them to move to their destinations through crowds or recommending other destinations. This reduces the management stress of practitioners and lowers the risk of virus transmission. Combining the location information of the number of persons on board with the value of the ship space capacity can calculate the maximum number of persons onboard based on the current capacity of the ship and the physical distance of quarantine regulations. Using the results of this study can help shipping companies make ticket plans.

Frequent and careful hand hygiene using water and soap or alcohol-based disinfectant solutions can mitigate the risk of COVID-19. It should be easy to access health promotion materials that promote the importance of handwashing facilities, alcohol-based hand

rubbing solutions, and hand hygiene and explain effective hand hygiene practices. The plan should include the availability of alcohol-based hand sanitizers in spaces expected to have people, such as restaurants, elevators, corridors, sanitary spaces, workplaces, and changing rooms in general entrance security inspection areas. Onboard positioning equipment, such as beacons, can be installed where hand washing and alcohol disinfectants are present. The passenger location information, location of the sink, and the time spent nearby can be used to calculate the number of times the crew cleaned their hands. By establishing a health database, it is possible to increase efficiency by paying attention to persons on board with a small number of clean lines and guiding them to wash their hands immediately to comply with quarantine regulations or disinfect them. In addition, the health examination protocol should be non-discriminatory. This research task technology provides basic location information of the number of persons on board, and a health check system can be established using it. If the distance between persons on board is compared and the area is entered and departed, the health of persons on board can be checked.

Before starting the voyage, the cruise ship operator must have a call route along the route and ensure that passengers and flight attendants can receive medical treatment if necessary. Repatriation and crew changes can be organized. If the confirmed case of COVID-19 is found in the board, the nearest port where testing for SARS-CoV-2 may occur, and the local public health authorities should be further managed, including special provisions. It does not matter, and if necessary, contact tracking is performed. The company should establish procedures to cope with potential COVID-19 outbreaks and establish programs for training and practice to prepare for such attacks. Crew members must receive the training necessary to perform the response mission. In particular, if there is a possibility of occurrence or confirmed case, all persons in charge of entering the quarantined area should be educated in terms of complying with all precautions. Training in preparation for this should be regularly organized on ships.

By the public health regulations of the destination country, passengers may request to create a hard copy of the passenger locator before boarding the cruise ship. Passenger locator data should be made available as soon as possible upon request so that public health authorities can initiate contact with exposed passengers. The rapid availability of accurate passenger locator data is critical to the success and effectiveness of contact tracking operations. The health authorities can identify and notify the contact information of infection cases to provide active follow-up management and related advice through this. The easiest way to obtain the passenger data needed for effective tracking will be direct cooperation between cruise companies, port authorities, and public health authorities.

### 6.2. Use of Development Technology: Rapid Action after Discovery of Confirmed Patients

High-level passenger location information obtained through onboard positioning technology can detect close contacts quickly. Firstly, the location information of the confirmed patient and the location information of the passenger are entered. Secondly, location information of the same type as the verified patient can be classified in DBSCAN through location information calculation. It is a technology that accurately identifies contacts within a short period and isolates them according to quarantine regulations based on the location information of confirmed patients and the OD data of passengers. Electronic fences can be installed when contacts are isolated. If a close contact leaves the quarantine area without permission, they are warned through a mobile phone voice trigger device. In short, our technology plan is effective in reducing the risk of virus transmission and can help customer-dependent industries (Figure 17) [28].

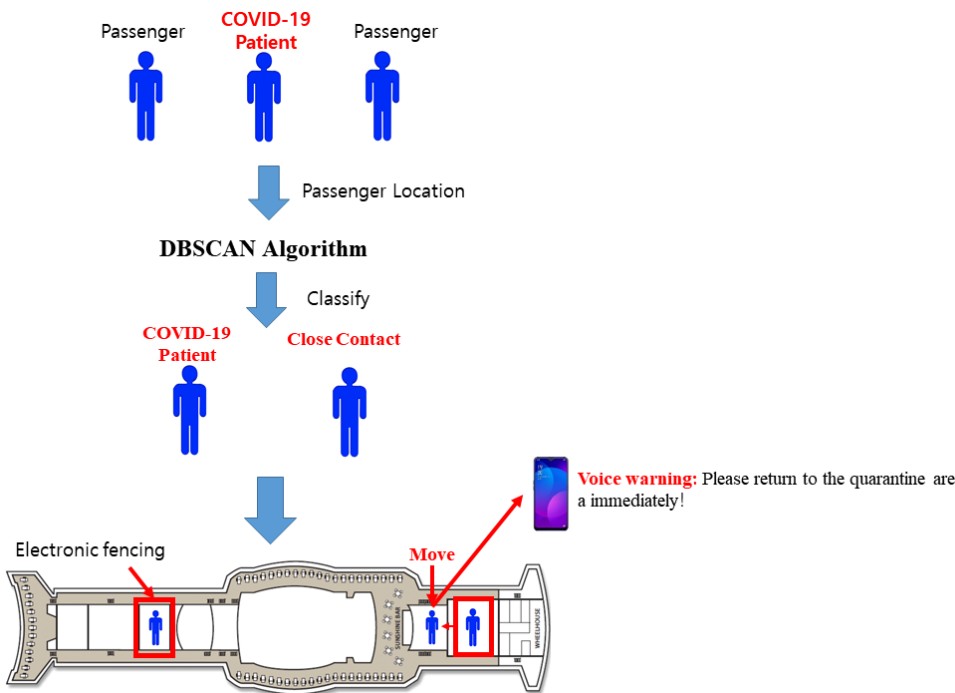

**Figure 17.** An example of applying development technology to the discovery of COVID-19 patients.

Contact tracking should be initiated by the health authorities who diagnosed the case and contacted the cruise ship company to quickly identify and contact passengers and crew exposed to the incident. Measures to support and promote such tracking may be as simple as asking passengers to provide contact details for follow-up if necessary. Collection of contact information should be performed electronically and ideally to facilitate and accelerate contacting people at risk and merge this information with a contact tracking database. Note that identifying a single confirmed case transmitted on a cruise would consider all passengers and crew members on board at the time to be high-risk contacts. Therefore, as described above, all passengers must contact and inform management, including quarantine, for 14 days after the last exposure. They should also be advised to seek further advice from the public health authorities to stay on follow-up measures. Movement route information on the number of persons on board can be obtained through onboard positioning technology. Once a person onboard is confirmed, close contacts can be quickly found, reducing the risk of spreading the virus and allowing workers to follow.

Tracking contacts is an essential measure to limit the spread of COVID-19. In general, it is performed only when the diagnostic results are accurate. Still, in cruise ship situations, it is recommended to start contact tracking already while waiting for the diagnostic results. Diagnostic data transmission and rapid testing on cruise ships are available, but they are not currently verified, and confirmation tests are only possible when anchored at the dock. Contact tracking should always be performed in cooperation with public health authorities. If a potential case is confirmed while the ship has not yet arrived at the port, the crew must begin tracking contact on board while checking input into the contact tracking process with public health authorities at the next port. All passengers should be evaluated for exposure and classified as high-risk or low-risk exposure. Passengers who meet the definition of a contact case must request information on the following. It will include contact with the place he visited, that is, a period of two days before symptoms appear. It is possible to quickly implement a contact or group tracking function based on onboard positioning technology.

The contact information of the likely cause should be managed as if the case had been confirmed until the final test results were obtained. If a possible issue is negative, no further action is required. The contract should be described below if the laboratory results are

positive. High-risk exposure contacts should be quarantined for 14 days after their last exposure to the case. Hygiene measures, respiratory etiquette should be strictly observed, symptoms should be monitored, ideally provided with a fever reducer, and what to do if symptoms appear. The quarantine facility is correct to be provided on land. Still, if not possible, the contact must stay in a specific quarantine room and provide food and other necessities while ensuring the crew's safety by providing such services. Crew members carry out cleaning at risk of exposure. A bathroom must be installed in the isolation room.

If two or more people share a room and only one is in high-risk contact, they must move to a single room. When two or more people share a room, if one person has symptoms, it should be managed as a possible cause, and the contact person should be subsequently accommodated in a separate room. After 14 days, the contact person must disembark safely and quarantine on land. Even if symptoms do not occur, the test should also be considered for high-risk exposure. Rapid identification of infection between contacts allows contact tracking to begin as soon as possible.

In-board positioning technology helps track the number of persons on board. This study plan identifies the areas where COVID-19 patients and close contacts stayed for the past 14 days. OD data of the occupant may be extracted using stop point recognition. Since OD data can appear anywhere on the ship, individual OD data is not meaningful. By classifying OD data, we can know which OD data is essential. Therefore, this function will be implemented using the DBSCAN algorithm. Based on this, the scope of the area where the number of persons on board with risk factors stayed is identified.

In this way, quarantine personnel can clarify the goal of disinfection rather than directly disinfecting the entire ship, thereby increasing the efficiency of quarantine work and making the most of existing resources. To realize this, the convex hull algorithm calculates the classified OD data, which could calculate the maximum boundaries of the points, thereby extracting the area where the occupant could stay (Figure 18) [28].

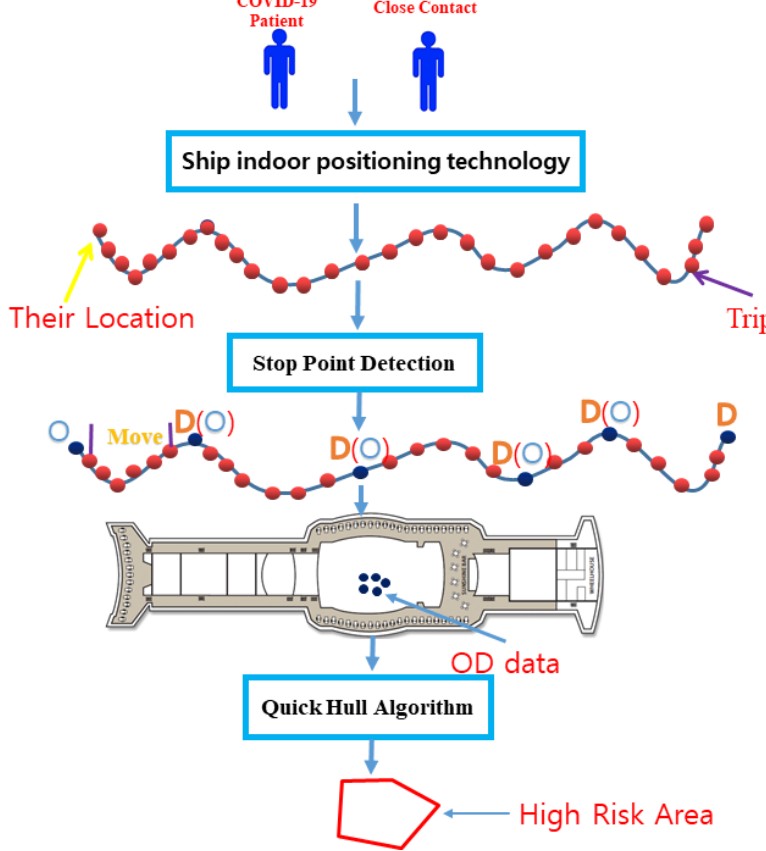

**Figure 18.** Identification of the activity area of the person in close contact with COVID-19.

The work of [31] presents their work on the indoor positioning of users, using a network of BLE beacons deployed in a large wholesale shopping store. The work of [32] proposes a Bluetooth location-based indoor positioning system for warehouse asset tracking to achieve a low-cost asset management solution. The work of [33] proposes an indoor positioning system to enhance the user experience in museums. The system relies on the proximity and localization capabilities of Bluetooth Low Energy beacons to automatically provide the user with cultural content related to the observed artwork. Evacuating people on board a ship is a typical use case for location services. For example, the traditional evacuation method uses lights, sounds, and signs in a fire onboard a ship. However, this can only be performed if the crew correctly identifies the signs and distinguishes the sounds. It is easy for the crew to lose their calm judgment in dangerous situations. Likely, they will not be able to identify the signs correctly. This is where location services are needed to guide the crew safely out of the situation in real-time.

Moreover, it is not only the crew who need to know where they are. The rescuers also need to know the position of the crew and their position. Location services are, therefore, essential in this case. Once the crew's location has been obtained, the shortest possible escape route can be planned quickly, reducing the likelihood of danger. In addition, the rescuers can pinpoint the location of a person in danger, thus reducing the rescue time. In this case, no information is required from the crew to rescue them. This is ideal for situations where a person is unconscious. Therefore, there is a vast potential for developing location services for ships.

Moreover, it is not only the crew who need to know where they are. The rescuers also need to see the position of the crew and their position. Location services are, therefore, essential in this case. Once the crew's location has been obtained, the shortest possible escape route can be planned quickly, reducing the likelihood of danger. In addition, the rescuers can pinpoint the location of a person in trouble, thus decreasing the rescue time. In this case, no information is required from the crew to rescue them. This is ideal for situations where a person is unconscious. Therefore, there is a vast potential for developing location services for ships.

## 7. Conclusions

The three main elements of sustainable ship management in the post-COVID-19 era are ship indoor positioning technology, close contact identification, and risk area calculation. In this paper, these three elements are discussed in detail. Firstly, it is about the ship's indoor positioning technology. There are complex signal reflection and scattering phenomena in the ship environment. This paper proposes the ship environment-aware indoor positioning algorithm for the mobile ship environment, achieving high accuracy localization without comparing fingerprint maps. The algorithm's accuracy is compared with three RSSI values: average, K-means, and peak. It is found that the peak value has the highest accuracy. Therefore, the peak value is used as the representative value of RSSI in this algorithm. In addition, this paper also compares two different indoor positioning algorithms, KNN and K-means RMSEs, of which are 2.04 and 1.79 m, respectively. The results show that the ship environment-aware algorithm has the highest accuracy. This algorithm can achieve high accuracy localization without fingerprint maps, and the RMSE is 1.63 m, which meets the localization requirements in the mobile ship environment.

The second is the identification of close contacts. Isolation of close contacts onboard ships is an effective way to reduce the risk of the virus spreading. The detection of close contacts has become one of the most critical issues for protecting the disease infection. The DBSCAN can identify close contacts based on their position similarity to patients. The patients and close contacts to them should be observed as one group. The shut-off spatial environment of ships facilitates the spread of the virus in a global COVID-19 pandemic environment. The most critical aspect of pandemic preparation is the timely detection of close contacts on board. The passenger location in the ship is based on an indoor positioning technology for ships. The last is the risk area calculation, the stop point detection to extract

the OD data. The DBSCAN algorithm is used to cluster passenger locations using the OD data. The convex hull algorithm is applied to calculate the boundary of the pressed passenger locations, which may be the hazardous areas. The visiting rate, the departure rate, and the average staying time of passengers are considered criteria to rank the risks for each identified area.

Finally, the hazardous areas are be measured by three metrics, and a list of places in the order of risk ranks is derived. This list will be handy in various ways, such as the early prevention of infectious diseases. For instance, the epidemic prevention personnel can use this list to disinfect accurately the identified hazardous areas one by one.

This paper presents the idea of sustainable management of ships with positioning services in the post-epidemic era. For the first time, a new technical framework is proposed to make sustainable management. It can be applied to different types of ships.

**Author Contributions:** Conceptualization, Q.L.; methodology, Q.L.; software, Q.L.; validation, Q.L.; formal analysis, Q.L.; investigation, Q.L.; resources, Q.L.; data curation, Q.L.; writing—original draft preparation, Q.L.; writing—review and editing, J.S.; visualization, Q.L.; supervision, J.S.; project administration, J.S.; funding acquisition, J.S. All authors have read and agreed to the published version of the manuscript.

**Funding:** This research was supported by Basic Science Research Program through the National Research Foundation of Korea (NRF), funded by the Ministry of Education (grant number: 2021R1I1A3056125).

**Institutional Review Board Statement:** Not applicable.

**Informed Consent Statement:** Not applicable.

**Data Availability Statement:** The data presented in this study are available on request from the corresponding author. The data are not publicly available due to security and privacy issues.

**Conflicts of Interest:** The authors declare no conflict of interest.

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
