# Peer review of "Sustainable Ship Management Post COVID-19 with In-Ship Positioning Services"

_sustainability, doi:10.3390/su14010369_

Round 1

Reviewer 1 Report

Dear Authors,

This manuscript has been reviewed, and some problems needed to be solved as follows.

1, how to determine parameters of DBSCAN?

2, how to validate the proposed method? What about the uncertainty of the proposed method?

3, why select DBSCAN? Is there a comparison with other similar methods?

4, is there some similar previous studies? What is the unique value of this study, please highlight.

Author Response

Response to Reviewer 1 Comments

Thanks very much for taking the time to review this manuscript. I really appreciate all your comments and suggestions! Please find my itemized responses in below.

Point 1: How to determine parameters of DBSCAN?

Response 1: DBSCAN has two parameters. One is Eps, and the other is MinPts. Eps is the radius. MinPts is the minimum number of points within the radius. Density is the measure by which DBSCAN clusters the points. Therefore, density is inscribed by Eps and MinPts. In general, Eps and MinPts are initialised. The clustering results of DBSCAN are usually shown in figures. If a result shown in the figure is not good, the two parameters should be manually adjusted to get a optimal clustering result. However, this method is not suitable for use in complex ship environments. This is because the ship environment may change at any time. In the shipping scenario, the space of the ship is relatively narrow. The behavioural ways of the persons on board are very different from those in a traditional indoor method. In addition, the boat has a long navigation period. The person on board cannot enter or leave the ship for some time, which is very different from the traditional indoor scenario. Most people live on land. As a result, there is a particular awareness of behavioural patterns indoors. This perception can help in the determination of Eps and MinPts parameters. However, most people have no experience living in ships and lack experience in the ship environment.

Faced with the many differences from traditional indoor positioning scenarios, it is difficult to determine these two parameters manually. Machine learning can give algorithms the ability to learn the parameters. However, the data first needs to be labelled. This is very labour- and material-intensive. In addition, there is very little positional data available in the shipping scenario. There is hardly any available data to use for labelling. As in this paper, DBSCAN is used to find close contacts. There are specific behavioural characteristics between close contacts and confirmed COVID-19 cases. According to the current definition of close contact, a person is considered comparable if they spend more than 15 minutes with a verified COVID-19 patient at a distance of 2 m or less due to the relatively small space of the ship. Therefore, the actual contact distance is less than 2 m. However, the exact value of the spread is difficult to determine. As a result, Eps is difficult to determine.

In general, the position data of the persons on board the ship needs to be obtained every second. Assume that the acquisition time is 15 minutes. Then there are 900 position data messages per person on board. In general, both close contacts and COVID-19 patients will be in the same position for each within 15 minutes. Therefore, it is assumed that one of the points of a confirmed COVID-19 patient is the core point. Within a range of Eps of 2, Minpts should be 900. However, given the error of indoor positioning and the different ship environments, Eps and MinPts are set to 2 and 450, respectively. After DBSCAN has been run, the clustering results for the user ID are returned. The prevention officer can confirm the clustering results. If the number of correct results is greater than the number of incorrect results, then only the wrong results are 0, and the rest are automatically marked as 1. If the number of correct results is less than the incorrect ones, then only the correct ones are marked as one, and the rest are automatically marked as 0.

A loss function for Eps, MinPts and labels is then constructed. This allows DBSCAN to adjust these two parameters continuously to minimise the loss value. In summary, the number of data labels is reduced by initialising Eps and MinPts. The construction of the loss function for these two parameters with the labels allows DBSCAN to adapt itself to the ship's environment and find the Eps and MinPts with the lowest loss.

This content is added in chapter 5, lines 417-459, pages 12-13.

Point 2: How to validate the proposed method? What about the uncertainty of the proposed plan?

Response 2: In this paper, we have proposed a practical plan (Sustainable ship management for post-COVID-19 with in-ship positioning services) and we have completed some of the experimental results. The complete experimental results are expected to be presented in a paper in the near future. For your information, we would like to describe how the next experiment will be done in more detail.

The approach proposed in this paper consists of three main techniques. They are the ship indoor positioning algorithm, close contactor identification and risk area identification. To validate the effectiveness of the proposed method, the location information of the user devices needs to be collected firstly. A deployment plan needs to be made based on the ship's floor plan to deploy the beacon devices accurately and efficiently. In the method proposed in this paper, the user device refers to a smartphone. Firstly, the RSSI data from the beacon needs to be collected using the smartphone's app. Secondly, the RSSI data is fed into the ship environment-aware indoor positioning algorithm to obtain the location information of the user device. The exact positioning process has been discussed in chapter 3. In the deployment planning, test locations for close contacts are pre-set. These test locations are located in different areas. There will be two or three test locations in different zones. In each area, the distance between the test locations is less than 2 m. At each test location, 15 minutes of RSSI data is collected using different smartphones.

To simulate a realistic scenario onboard the ship, a person on board with a smartphone will walk through the area where these test locations are located. The test positions are also estimated using the ship environment-aware indoor positioning algorithm. The resulting result of location estimation is fed into the close contact identification algorithm, which returns a clustering result. The categories to which the user IDs of close contacts and COVID-19 confirmers belong in the deployment planning are known. The estimated results can be compared to the known results to verify that the close contact identification algorithm is valid. Finally, the risk area identification algorithm is used to calculate the area where the estimated test location is located. Since the area in which the test location is located is known in the deployment planning. Therefore, the similarities and differences between the estimated size and the known area can be compared to verify the validity of the risk area identification algorithm.

The uncertainty about the proposed method mainly comes from the localisation error. Traditional indoor localisation algorithms use fingerprint maps. However, in a complex environment, the fingerprint map is constantly changing. Fingerprint map needs to be recreated at regular intervals, which is difficult to achieve in a mobile environment like a ship. Therefore, the ship environment-aware indoor positioning algorithm proposed in this paper does not rely on a fingerprint map for positioning. Thus, the uncertainty of the proposed method is reduced.

This content has been added in chapter 2, lines 187-218, pages 4-5.

Point 3: Why select DBSCAN? Is there a comparison with other similar methods?

Response 3: In a real ship scenario, it is not possible to determine the number of close contacts and the number of categories of confirmed COVID-19 patients. In other words, the number of infections and the number of close connections is unknown. Therefore, the classification algorithm is not suitable for application in this scenario. Clustering algorithms do not need to know the number of clusters. There are currently three different approaches to clustering data. They are Partition-based Methods, Hierarchical Methods and Density-based methods [29]. Firstly, among the Partition-based Methods, the most representative one is the K-means algorithm, which requires the determination of the number of clusters or cluster centres, i.e. the pre-specification of K. The core of this algorithm is to make the points in a cluster close enough and the points between clusters far enough. Secondly, in Hierarchical Methods, there are two general approaches. One is Agglomerative hierarchical clustering, and the other is Divisive hierarchical clustering. In this example, Agglomerative hierarchical clustering treats each point as a cluster, and by merging them, a cluster is obtained. But Hierarchical Methods also needs to specify the number of clusters.

Finally, among the Density-based methods, DBSCAN is the most representative algorithm, which uses the concept of density to cluster the data. Density differentiation is most appropriate in the identification of close contacts. From a positional point of view, close contacts and confirmed COVID-19 diagnosed persons have a high similarity of characteristics in terms of density. This also means that clustering by density has high accuracy and efficiency. Not only that, but density-based clustering algorithms do not require a specified number of clusters. This makes it difficult to correctly determine K when the number of infected persons and close contacts is unknown. Both require storage of the cluster relationships between points, which poses a significant challenge in algorithmic efficiency and storage space.

Compared to other clustering methods, Figure 10 shows the comparison results of different clustering algorithms. The latitude and longitude ranges of (a) to (d) in Figure 10 are the same. (a) is the correct classification result. The right number of classification results is 4. The number of K-means and Hierarchical algorithms clusters is directly specified as 4. It can be seen from the red box range that the clustering result of DBSCAN is closest to the correct clustering result. Therefore, DBSCAN is applied in the study of this paper.

This content has been added in chapter 5, lines 379-408, page 11.

Point 4: Is there some similar previous studies? What is the unique value of this study, please highlight.

Response 4: As your comment, we have highlighted our contribution in the last paragraph in the sections of Introduction (chapter 2, lines 78-105, pages 2-3) and Conclusion (chapter 7, lines 784-786, pages 23). The content added is as follows: Ship indoor positioning algorithms, close contact identification algorithms and risk area identification algorithms have been studied similarly in their respective fields. However, there are no examples of close contact identification algorithms and risk area identification algorithms studied in a ship environment. The topic presented in this paper is location services for the sustainable management of ships. The ship environment-aware indoor positioning algorithm proposed in this paper is applicable in the ship environment. Unlike traditional indoor positioning algorithms, the ship environment-aware indoor positioning algorithm does not require the creation of a fingerprint map. The nearest beacon is determined based on the peak value of the RSSI collected by the user device. The ship environment changes frequently and has a more significant impact on the fingerprint profile. Therefore, this approach gets rid of the dependence on fingerprint maps. Thereby the algorithm can be adapted to the ship environment.

In the post-epidemic era, identifying close contacts is essential for management. The identification of close contacts is also an application case for location services. But the ship environment is very different from a typical indoor environment. The process and algorithm details for identifying close contacts based on location are also other. Close contact recognition algorithms need to be studied for unique environments such as ships. For now, this paper fills the research gap of close contact identification algorithms in the context of ships. In addition, it is the first time that a risk area identification algorithm has been proposed for the ship environment. During a ship's voyage, the persons on board cannot enter or leave the boat. Then in a closed environment like a ship, the persons on board will gather in some areas. Therefore, in the post-epidemic era, these areas are at risk. However, the formation of these areas depends on the behavioural patterns of the people travelling on board. Due to the specificity of the ship environment, it is necessary to research risk area identification algorithms in the context of ships. Overall, this paper presents the idea of sustainable management of ships with positioning services in the post-epidemic era. And for the first time, a new technical framework is proposed to make practically sustainable management. It can also be applied to different types of ships. 

Reviewer 2 Report

The paper deals with a very current topic related to ship management with reference to limiting the on-board spread of pandemics. Specifically, an algorithm for indoor positioning is presented with a rigorous scientific methodology. However, it can improved. Following some suggestions:

Improve Figure2: better to represent it with a block diagram

Enlarge Figure 15.

Add some specific reference about the risk-management on board ships during pandemics after line 36.

In line 478 and similar, use “persons on board” instead of “people on board”.

The positioning algorithms of persons based on beacons can also find other applications on board ships, for example providing a useful aid during the evacuation process. Could the authors add some consideration about it?

Author Response

Response to Reviewer 2 Comments

Thank you very much for your careful review and constructive suggestions with regard to our manuscript. The main corrections in the paper and the responses to the reviewer comments are as follows:

Point 1: Improve Figure2: better to represent it with a block diagram.

Response 1: Figure 2 has been refined to express it as a block diagram.

Point 2: Enlarge Figure 15.

Response 2: Figure 15 has been enlarged

Point 3: Add some specific reference about the risk-management on board ships during pandemics after line 36.

Response 3: We have added some specific references about the risk management on board ships. Risk management is the most crucial measure to ensure the proper operation. This content has been added to chapter 1, lines 37-45, pages 1-2. The content added is as follows: “[30] investigates the frequency, circumstances, and causes of occupational accidents on merchant ships. They conclude that occupational accidents remain a crucial issue for the shipping industry. [31] identifies key points for accident prevention on board ships, with people being critical points. Threat information collected and recorded using a structured approach can improve ship safety by selecting risk control options [32]. [33] considers risk management of ship fire incidents as an essential issue in maritime transport systems. Ultimately, risk management is really about managing people. Therefore, real-time access to information on the location of persons on board is an essential element of risk management”.

Point 4: In line 478 and similar, use "persons on board" instead of "people on board."

Response 4: The phrase "people on board" has been replaced with "persons on board" throughout the text.

Point 5: The positioning algorithms of persons based on beacons can also find other applications on board ships, for example providing a useful aid during the evacuation process. Could the authors add some consideration about it?

Response 5: We have added more cases as you have commented. This content has been added to chapter 6, lines 719-739, page 21. The content added is as follows: “[34] presents their work on the indoor positioning of users, using a network of BLE beacons deployed in a large wholesale shopping store. [35] proposes a Bluetooth location-based indoor positioning system for warehouse asset tracking to achieve a low-cost asset management solution. [36] proposes an indoor positioning system to enhance the user experience in museums. The system relies on the proximity and localization capabilities of Bluetooth Low Energy beacons to automatically provide the user with cultural content related to the observed artwork. Evacuating people on board a ship is a typical use case for location services. For example, the traditional evacuation method uses lights, sounds, and signs in a fire on board a ship. However, this can only be done if the crew correctly identifies the signs and distinguishes the sounds. It is easy for the crew to lose their calm judgment in dangerous situations. Likely, they will not be able to identify the signs correctly. This is where location services are needed to guide the crew safely out of the situation in real-time.

Moreover, it is not only the crew who need to know where they are. The rescuers also need to know the position of the crew and their position. Location services are, therefore, essential in this case. Once the crew's location has been obtained, the shortest possible escape route can be planned quickly, reducing the likelihood of danger. In addition, the rescuers can pinpoint the location of a person in danger, thus reducing the rescue time. In this case, no information is required from the crew to rescue them. This is ideal for situations where a person is unconscious. Therefore, there is a vast potential for developing location services for ships”. 

Reviewer 3 Report

The article Sustainable ship management for post-COVID-19 with in-ship positioning services is an original work. The authors described the value of their research in the words: … this paper proposes for the first time that three technologies are needed to support the sustainable management of ships in the post-COVID-19 era. They are ship indoor positioning, close contact identification, and risk area calculation. The Authors present a solution that is supported by extensive analyses. This is further supplemented in section 2 Previous Work with the following words:  … this paper proposes a new algorithm that does not rely on the fingerprint profile in the offline phase. The localization is performed directly in the online phase and high accuracy localization results can be obtained. The new indoor positioning is called environment-aware indoor positioning algorithms. The article refers to a currently important problem not only on land but also on-board ships.

In my review, I refer to the formal page of the article and mainly editorial side. I do not engage in polemics and discussions on the considerations adopted by the Authors because it determines the originality of the issue in the article. Considering the proposed solutions, I accept it with appreciation.

The following observation require to draw attention and response from the Authors:

  1. The introduction section does not include the composition and content of individual sections of the article, which is usually the case in the last sentences of this section. This description of the content of the article is used in MDPI publications. (Introduction Section).
  2. Figure 15 is too small and therefore not legible. The reviewer suggests enlarging this drawing to make it adequately accurate and comparable with the integrity of the other figures (Line 393).
  3. Part of the above remark applies to Figures 16 (Line 420), Figure 17 (Line 545) and Figure 18 (Line 608).
  4. Line 546 seems to be unnecessary.

In my review, I do not include more comments other than those presented in the review form. In the article, the Authors have used 28 items of references. Most of the publications included in the References are contemporary publications for obvious epidemic reasons. The text of the article is clear.

Author Response

Response to Reviewer 3 Comments

Thanks very much for taking the time to review this manuscript. I appreciate all your comments and suggestions! Thank you very much for your recognition of our paper. Please find my itemized responses below and my revisions in the re-submitted files.

Point 1: The introduction section does not include the composition and content of individual sections of the article, which is usually the case in the last sentences of this section. This description of the content of the article is used in MDPI publications. (Introduction Section).

Response 1: We have added some more sentences as you have commented. The sentences have been added in chapter 1, lines 112-121, and page 3 as follows: This paper is organized as Chapter 3 introduces the ship environment-aware indoor positioning algorithm to obtain the location information of the people riding the ship and to compare other different indoor positioning algorithms. Chapter 4 introduces the method of mobile information extraction. Moreover, the mobile information extraction is based on the location information of the ship's crew. In addition, the concept of area of interest is introduced to calculate the risk area. Chapter 5 is to introduce close contact person identification and risk area identification. The close contacts are distinguished by the clustering algorithm. After the clustering results are obtained, the risk region calculation is then performed. Chapter 6 is a discussion on how to manage ships sustainably in the post-epidemic era based on our proposed approach.

Point 2: Figure 15 is too small and therefore not legible. The reviewer suggests enlarging this drawing to make it adequately accurate and comparable with the integrity of the other figures (Line 393).

Response 2: Figure 15 has been enlarged.

Point 3: Part of the above remark applies to Figures 16 (Line 420), Figure 17 (Line 545) and Figure 18 (Line 608).

Response 3: Figure 16, Figure 17, and Figure 18 have been enlarged.

Point 4: Line 546 seems to be unnecessary.

Response 4: Line 546 has been removed.

Round 2

Reviewer 1 Report

Dear Editor,

The current form can be accepted for this journal.